# Loss of family with sequence similarity 13, member A exacerbates pulmonary hypertension through accelerating endothelial-to-mesenchymal transition

Pranindya Rinastiti[1,2], Koji Ikeda[1]*, Elda Putri Rahardini[1,2], Kazuya Miyagawa[1,2], Naoki Tamada[1,2], Yuko Kuribayashi[1], Ken-ichi Hirata[2], Noriaki Emoto[1,2]

**1** Laboratory of Clinical Pharmaceutical Science, Kobe Pharmaceutical University, Kobe, Japan, **2** Division of Cardiovascular Medicine, Department of Internal Medicine, Kobe University Graduate School of Medicine, Kobe, Japan

* ikedak-circ@umin.ac.jp

## Abstract

Pulmonary hypertension is a progressive lung disease with poor prognosis due to the consequent right heart ventricular failure. Pulmonary artery remodeling and dysfunction are culprits for pathologically increased pulmonary arterial pressure, but their underlying molecular mechanisms remain to be elucidated. Previous genome-wide association studies revealed a significant correlation between the genetic locus of family with sequence similarity 13, member A (*FAM13A*) and various lung diseases such as chronic obstructive pulmonary disease and pulmonary fibrosis; however whether *FAM13A* is also involved in the pathogenesis of pulmonary hypertension remained unknown. Here, we identified a significant role of *FAM13A* in the development of pulmonary hypertension. *FAM13A* expression was reduced in the lungs of mice with hypoxia-induced pulmonary hypertension. We identified that *FAM13A* was expressed in lung vasculatures, especially in endothelial cells. Genetic loss of *FAM13A* exacerbated pulmonary hypertension in mice exposed to chronic hypoxia in association with deteriorated pulmonary artery remodeling. Mechanistically, *FAM13A* decelerated endothelial-to-mesenchymal transition potentially by inhibiting β-catenin signaling in pulmonary artery endothelial cells. Our data revealed a protective role of *FAM13A* in the development of pulmonary hypertension, and therefore increasing and/or preserving *FAM13A* expression in pulmonary artery endothelial cells is an attractive therapeutic strategy for the treatment of pulmonary hypertension.

**Data Availability Statement:** All relevant data are within the manuscript and its Supporting Information files.

## Introduction

Pulmonary hypertension is a progressive and fatal lung disease diagnosed by a sustained elevation of pulmonary arterial pressure more than 20 mmHg [1]. Pulmonary arterial hypertension including idiopathic pulmonary arterial hypertension and pulmonary hypertension related with collagen disease is characterized by pathological pulmonary artery remodeling such as intimal and medial thickening of muscular arteries, vaso-occlusive lesions, and fully

**Funding:** The authors receive no specific funding for this work.

**Competing interests:** The authors have declared that no competing interests exist.

muscularized small diameter vessels that are normally non-muscular peripheral vessels. These vascular remodeling is a result from endothelial cell dysfunction, smooth muscle cell and endothelial cell proliferation, and also cellular transdifferentiation [2]. Although detailed molecular mechanisms remain to be elucidated, many pathogenic pathways in pulmonary arterial hypertension have been revealed. These include TGF-β signaling, inflammation, pericyte-mediated vascular remodeling, iron homeostasis, and endothelial-to-mesenchymal transition (EndMT) [3].

Recent genome-wide association studies identified family with sequence similarity 13, member A (*FAM13A*) gene as a genetic locus associated with pulmonary function [4], and it is known to be associated with lung diseases including chronic obstructive pulmonary disease (COPD) [5], asthma [6] and pulmonary fibrosis [7–9]. Moreover, causative role of *FAM13A* in the development of COPD has been revealed. *FAM13A* interacts with protein phosphatase 2A and β-catenin, leading to the promotion of GSK-3β-mediated phosphorylation and subsequent proteasomal degradation of β-catenin in airway epithelial cells [10]. Interestingly, *FAM13A* is also expressed in adipocytes, and modulates insulin signaling through regulating the proteasomal degradation of insulin receptor substrate-1 [11].

β-catenin is crucially involved in the epithelial-mesenchymal transition that plays an important role in the pathogenesis of cancer [12] and pulmonary fibrosis. Also, there are many reports describing the role of β-catenin in EndMT that is implicated in the vascular remodeling for pulmonary hypertension [13–15]. These findings urged us to investigate a potential role of *FAM13A* in the pathogenesis of pulmonary hypertension, and we here identify a protective role of *FAM13A* in the development of pulmonary hypertension.

## Materials and methods

### Animal study

All animal experimental protocols were approved by Ethics Review Committee for Animal Experimentation of Kobe Pharmaceutical University. *Fam13a*$^{-/-}$ mice [*Fam13a* tm1e(KOMP) Wtsi; C57BL6N background] in which LacZ cassette was knocked in at the *Fam13a* gene locus were obtained from Knockout Mouse Project (KOMP) at UC Davis. Mice were maintained under standard conditions with free access to food and water.

Mice at 6–7 weeks old were regularly used for experiments. For chronic hypoxia exposure, mice were put in the chamber with non-recirculating gas mixture of 10% $O_2$ and 90% $N_2$ for 3–6 weeks. When sacrifice the mice, mice were anesthetized with ~2% isoflurane inhalation, followed by cervical dislocation.

### Hemodynamic measurements

Mice were anesthetized with ~2% isoflurane, and right ventricular systolic pressure was measured by inserting 1.4 F Millar Mikro-Tip catheter transducer (Millar) into right ventricle through right jugular vein. Before the hemodynamic assessments, heart rate, fractional shortening, cardiac output, and pulmonary artery acceleration time were evaluated by echocardiography.

### Right ventricular hypertrophy assessment

Formaldehyde-fixed dried hearts were dissected, and right ventricular wall were separated from left ventricle and septum. The Fulton's index was presented in ratio of right ventricle to left ventricle + septum.

## Histological analysis

Mouse lungs were inflated and fixed in 4% paraformaldehyde, followed by paraffin embedding. Sections were cut into 3 μm and stained with hematoxylin and eosin (HE) as well as Elastica van Gieson (EvG). Pulmonary artery wall thickness was assessed in HE-stained lung sections using imageJ by measuring 10 randomly selected vessels/mouse associated with alveolar duct or alveolar wall, with diameter less than 100 μm in 200x magnification. Quantitative data were presented as the wall area measurement (vessel area minus lumen area) normalized to the mean of vessel and lumen perimeters. Small pulmonary arteries number was evaluated in EvG-stained lung sections. Five fields were taken per mouse at 200x magnification and the number of distal arteries <50 μm in diameter per 100 alveoli were assessed.

To assess small pulmonary arteries muscularization, lung sections were incubated with Antigen Unmasking Solution (Citric-acid based) H-3300 (Vector Laboratories) at 90°C for 10 min, followed by incubation in PBS/0.2% Triton X-100 and subsequent blocking with 5% skim-milk for 1 h. Sections were then incubated with antibodies for α-smooth muscle actin (1:300; Sigma) and von Willebrand factor (vWF) (1:300; Abcam) at 4°C overnight. Subsequently, sections were incubated with secondary antibody labeled with Alexa Fluor 594 (1:300; Invitrogen), followed by mounting with Vectashield mounting medium with DAPI (Vector Laboratories). Fluorescent images were captured using fluorescence microscope (BZ-X800, Keyence). Small pulmonary artery with diameter less than 50 μm were quantified from 5 random fields at 400x magnification per mouse, and arteries with positive α-smooth muscle actin staining >75% of the circumference were classified as fully muscularized as previously described [16]. Data were presented as percentage of fully muscularized vessels normalized with total number of vessels per field.

For some immunostaining experiments, images were captured using laser confocal microscope (LSM700, Zeiss). For the assessment of colocalization, 15–20 randomly selected vessels (diameter size <50 μm) were analyzed for each group, and quantification was performed using Zen imaging software (Zeiss). Data were presented in Manders overlap coefficient as previously described [17]. To assess the nuclear accumulation of active β-catenin, >50 nuclei per group were analyzed by measuring the mean fluorescence intensity of active β-catenin using imageJ as previously described [12].

## LacZ staining

Right lung was flushed with 0.2% glutaraldehyde in wash buffer (2 mM MgCl$_2$, 0.01% Deoxycholate, 0.02% NP-40), and incubated in wash buffer on ice for 40 min. Lung samples were then washed with wash buffer for 30 min 3 times, and then cut into pieces. Lung specimens were then incubated in X-gal staining solution (5 mM K$_4$Fe(CN)$_6$, 5 mM K$_3$Fe(CN)$_6$, 0,75 mg/ml X-gal, diluted in wash buffer) overnight at 37°C on the rotator. After washing with 3% DMSO/PBS for 5 min twice, specimens were post-fixed with 4% PFA overnight at 4°C. Subsequently, lung specimens were embedded with paraffin, and sections were prepared at 3 μm, followed by counterstaining with Nuclear Fast Red Solution.

Immunostaining for LacZ and endothelial cells was performed using frozen sections of the right lung by incubating with antibodies for β-galactosidase (1:1000; Abcam) and vWF (1:250; Abcam), followed by incubation with secondary antibodies labeled with Alexa Fluor 594 (1:500; Abcam) and Alexa Fluor 488 (1:500; Invitrogen).

## Quantitative real time-PCR

Left lung tissues were collected and homogenized in RNAiso plus (TAKARA), followed by purification with Nucleospin RNA clean-up (Macherey-Nagel). cDNA was synthesized using

PrimeScript RT reagent Kit with gDNA Eraser (TAKARA). PCR reactions were prepared using FastStart SYBR Green Master (Roche Applied Science), followed by the real-time PCR analysis using LightCycler96 (Roche Applied Science). Nucleotide sequences of the primers are shown in S1 Table.

## Immunoblotting

Cells were lysed in RIPA buffer, and protein concentration was measured using DC Protein Assay Kit (BioRad). Proteins of 20–30 μg were run on the SDS-PAGE gel, and the expression of target proteins were assessed by immunoblotting. Target protein expression levels were normalized to GAPDH expression levels. Primary antibodies used were as follows: FAM13A (Sigma, HPA038108), transgelin (TAGLN, Abcam, #ab14106), Snail (Cell Signaling Technology, C15D3), PECAM-1 (Santa Cruz, #sc-1506), caspase-3 (Cell Signaling Technology, 8G10), cleaved caspase-3 (Cell Signaling Technology, D175), β-catenin (Cell Signaling Technology, #9562), active β-catenin (Cell Signaling Technology, D13A1), GAPDH (Cell Signaling Technology, 14C10). All antibodies were used at 1:1000 dilution unless otherwise mentioned.

## Cell culture

Human pulmonary artery endothelial cells (PAECs) were obtained from Lonza, and cultured in Humedia-EG2 (Kurabo). To induce endothelial-to-mesenchymal transition, PAECs were treated with 10 ng/mL TGF-β1 (R&D System) and 10 ng/mL IL-1β (R&D System) as previously described [18] for 6 days in the medium supplemented with 2% FBS. Medium was changed every other day during the experiments.

For retrovirus infection, PAECs were grown to 70% confluency, and incubated with a medium containing retrovirus carrying GFP or *FAM13A* gene in the presence of polybrene (8 μg/mL) for 24 h. The medium was then replaced with a fresh growth medium, and the cells were treated or used for experiments 48 h after initial infection. In some experiments, PAECs were transfected with either negative (Ambion, Silencer Select Negative Control siRNA) or Fam13a (Dharmacon, #M-020516-02-0005) siRNA using RNAiMAX transfection reagent (Thermo). Subsequently, EndMT was induced using 10 ng/mL TGF-β1 and 10 ng/mL IL-1β, followed by immunocytochemistry for PECAM-1 and α-smooth muscle actin (αSMA; Sigma, #F3777).

## Tube-formation assay

PAECs were plated in 96-well plate coated with 50 μl of Matrigel (Corning) with seeding density of $2 \times 10^4$ cells/well. The cells were incubated for 7 h, and images were obtained every hour.

## Cell proliferation assay

PAECs were plated in 96-well plate at the density of $0.1 \times 10^4$ cells/well. The medium was replaced after 24 h. Cell proliferation was assessed using WST-1 (Roche) at 48 h after seeding.

## Apoptosis assay

PAECs were plated in 96-well plate at the density of $1 \times 10^4$ cells/well, and incubated overnight. Apoptosis was induced by serum and growth factor depletion for 24 h, followed by TUNEL staining using In Situ Cell Death Detection Kit (Roche) or immunoblotting for cleaved caspase-3.

## Migration assay

Cell migration was analyzed using modified Boyden Chamber Assay. Cells ($8 \times 10^4$ cells/insert) were seeded on the 8 μm-pore insert (Falcon) in migration buffer (serum free medium supplemented with 1% bovine serum albumin). Migration was induced by 3% FBS in the migration buffer added in the bottom chamber. After 4 h of incubation, non-migrated cells on the top of insert membrane were removed, and the cells migrated onto the reverse face of the membrane were fixed in methanol, and stained with Hematoxylin.

## Statistical analysis

All data are presented as the mean ± SEM. Statistical analysis was performed using Graphpad Prism 8. The differences between groups were calculated using two-tailed Student's *t*-test or one-way ANOVA, as indicated. P < 0.05 was considered statistically significant.

## Results

### *FAM13A* is expressed in lung vasculatures

To assess a possible involvement of *FAM13A* in the pathogenesis of pulmonary hypertension, we analyzed *Fam13a* expression in the lungs of mice with pulmonary hypertension. Pulmonary hypertension was induced in mice by chronic exposure to 10% hypoxia. *Fam13a* expression was remarkably reduced in the lungs of wild-type (WT) mice exposed to chronic hypoxia comparing to that in the lungs of control mice under normoxic condition (Fig 1A and 1B). It has been reported that *FAM13A* is expressed in airway cells including mucosal cells, Club cells, and alveolar cells [8,10]; however it remained unknown whether *FAM13A* is expressed in the lung vasculature as well. We have generated mice with target deletion of *FAM13A* (*Fam13a*⁻/⁻) in which LacZ cassette was inserted into the intron of the *Fam13a* gene locus. We could therefore detect cells that express *Fam13a* by using LacZ staining in the lungs. LacZ-staining was positive in airway epithelial cells as previously reported, and some of vascular cells were also positive for LacZ in the lung of *Fam13a*⁻/⁻ mice under normoxic condition (Fig 1C). Immunohistochemistry using antibodies for vWF and LacZ demonstrated that vWF-positive endothelial cells express *Fam13a* in lung vasculatures of mice under normoxic condition (Fig 1D). These data suggest a potential role of *FAM13A* in the pathogenesis of pulmonary hypertension, especially in the pathological vascular remodeling.

### Genetic loss of *FAM13A* exacerbates pulmonary hypertension

We then explored a role of *FAM13A* in pulmonary hypertension using *Fam13a*⁻/⁻ mice. Under normoxic condition, there was no significant difference in lung structures, pulmonary arterial pressure, and hemodynamics between WT and *Fam13a*⁻/⁻ mice (Figs 2A–2C and 3A and 3B). When exposed to chronic hypoxia, *Fam13a*⁻/⁻ mice showed deteriorated pulmonary hypertension assessed by higher right ventricular systolic pressure and increased Fulton index (Fig 3A–3C). Consistent with the exacerbated pulmonary hypertension, vascular remodeling such as increased fully muscularized small diameter vessels and loss of peripheral capillaries was worsened in the lungs of *Fam13a*⁻/⁻ mice as compared to those in WT mice (Fig 3D–3F). These data sufficiently indicate that *FAM13A* plays a protective role against pulmonary hypertension.

### Loss of *FAM13A* promotes EndMT

Because *FAM13A* plays a crucial role in the regulation of β-catenin signaling that is involved in EndMT, we assessed the EndMT process in the lungs of WT and *Fam13a*⁻/⁻ mice exposed to chronic hypoxia. Expression of endothelial markers such as *Pecam1* and *Cdh5* was reduced,

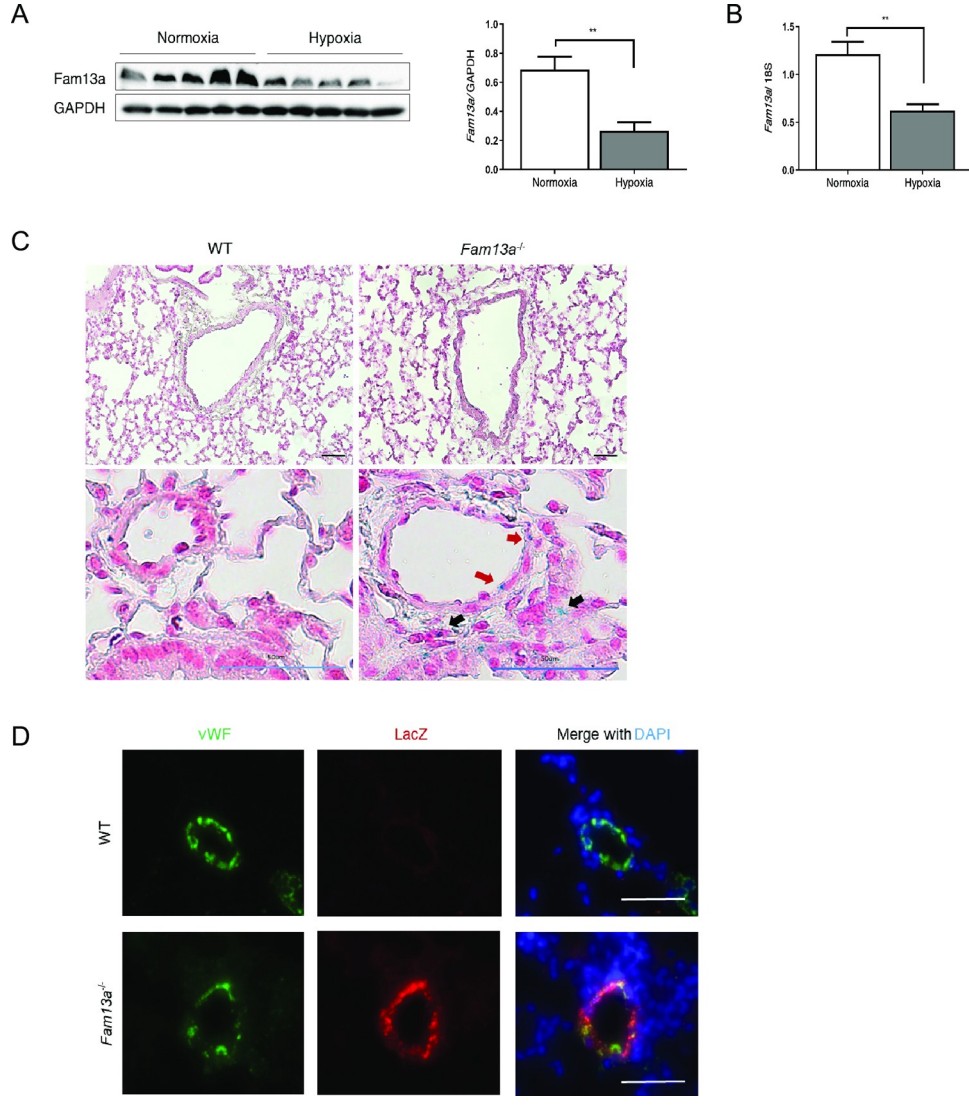

**Fig 1. *FAM13A* is expressed in lung vasculatures.** (A) Immunoblotting for *Fam13a* in the lungs isolated from WT mice exposed to either normoxic or hypoxic (10% $O_2$ for 6 weeks) conditions. (n = 5 each). (B) Quantitative real-time PCR of *Fam13a* in the lungs isolated from WT mice exposed to either normoxic or hypoxic conditions (n = 5 each). (C) Representative images of the lung sections stained with X-gal in WT and *Fam13a⁻/⁻* mice under normoxic condition. LacZ-positive cells that are supposed to express *Fam13a* were detected in pulmonary arteries (red arrow) as well as in non-vessel structures (black arrow). Bars: 50 μm. (D) Immunohistochemistry for LacZ and vWF in the lung of *Fam13a⁻/⁻* mice under normoxic condition. Bars: 50 μm. Data are presented as the mean ± SEM. *P < 0.05 and **P < 0.01.

while mesenchymal markers including *Acta1* and *Fn1* were significantly increased in the lungs of *Fam13a⁻/⁻* mice comparing to those in the lungs of WT mice (Fig 4A). Transcription factors such as *Twist1* and *Snai1* showed a marked increase in the lungs of *Fam13a⁻/⁻* mice as well (Fig 4A). Enhanced expression of mesenchymal markers in the lungs of *Fam13a⁻/⁻* mice was also confirmed by immunoblotting (Fig 4B). Furthermore, EndMT assessed by the emergence of endothelial and mesenchymal marker-double positive cells was apparently enhanced in the lungs of *Fam13a⁻/⁻* mice compared to that in WT mice (Fig 4C and 4D). These data suggest that loss of *FAM13A* promotes EndMT, resulting in the deteriorated pulmonary vascular remodeling and consequent pulmonary hypertension.

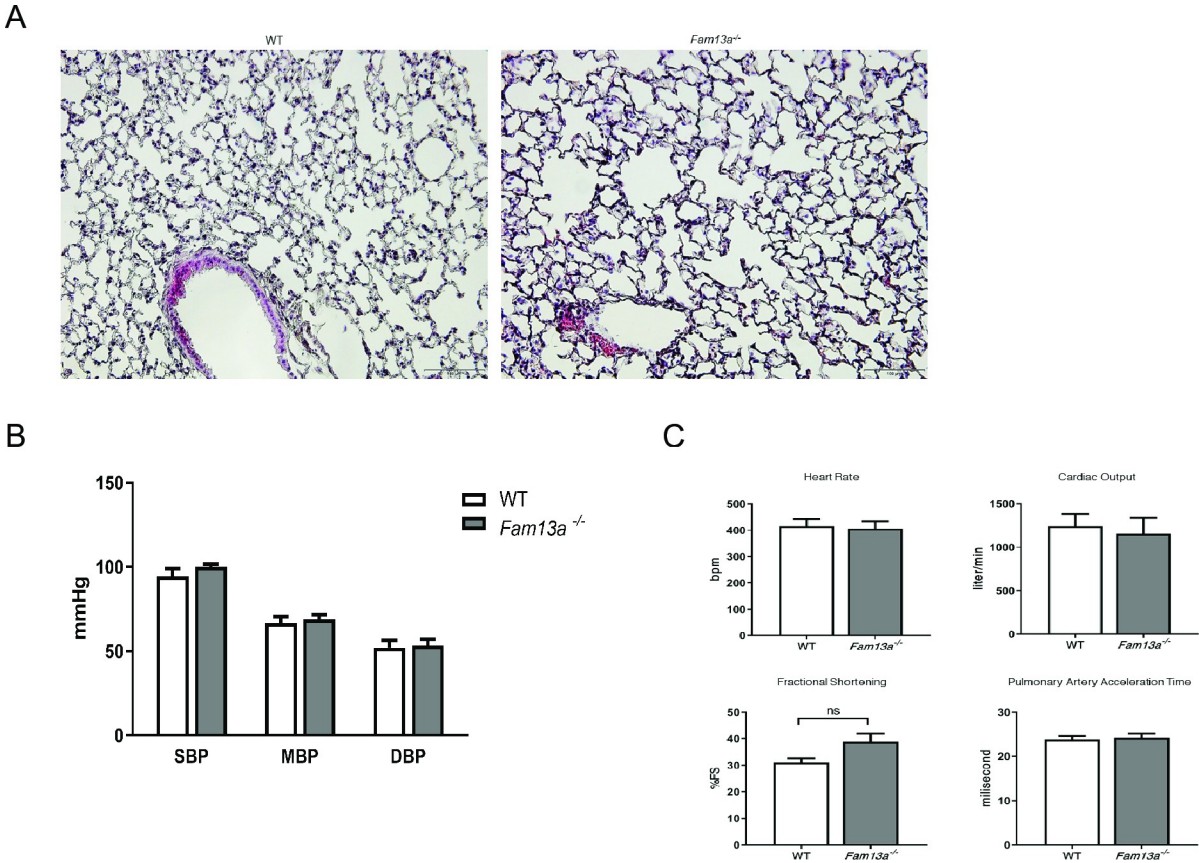

**Fig 2. Genetic loss of *FAM13A* does not affect the lung structures and hemodynamics in mice.** (A) HE staining of the lung sections isolated from WT and *Fam13a* $^{-/-}$ mice under normoxic condition. (B) Blood pressure in WT and *Fam13a* $^{-/-}$ mice under normoxic condition (n = 5–6 each). (C) Hemodynamics assessed by echocardiography in WT and *Fam13a* $^{-/-}$ mice under normoxic condition (n = 5 each).

## *FAM13A* reduces active β-catenin in endothelial cells, and decelerates the EndMT

We explored a role of *FAM13A* in EndMT using human pulmonary artery endothelial cells (PAECs) *in vitro*. When EndMT was induced by IL-1β and TGF-β1 treatment, *FAM13A* expression was significantly reduced in PAECs (Fig 5A). We then overexpressed *FAM13A* in PAECs using retrovirus-mediated gene transfection, and subsequently treated cells with IL-1β and TGF-β1 to induced EndMT. Overexpression of *FAM13A* inhibited the induction of mesenchymal markers, whereas reduction of endothelial markers was not affected (Fig 5B). In contrast, knockdown of Fam13a using siRNA appeared to enhance the loss of endothelial marker expression in PAECs undergoing EndMT assessed by immunocytochemistry, while mesenchymal marker expression was not detectable even after the EndMT-induction in both groups (Fig 5C). These data strongly suggest an inhibitory role of *FAM13A* in the EndMT process. In contrast, endothelial angiogenic capacities such as tube-formation, migration, proliferation, and apoptosis were not affected by *FAM13A*-overexpression in PAECs (Fig 5D–5H).

We then examined whether *FAM13A* modifies β-catenin signaling in PAECs. Overexpression of *FAM13A* significantly reduced the non-phosphorylated active β-catenin after EndMT induction, while total β-catenin protein levels did not change in PAECs (Fig 5I). Furthermore, nuclear accumulation of active β-catenin was significantly reduced in PAECs that overexpress

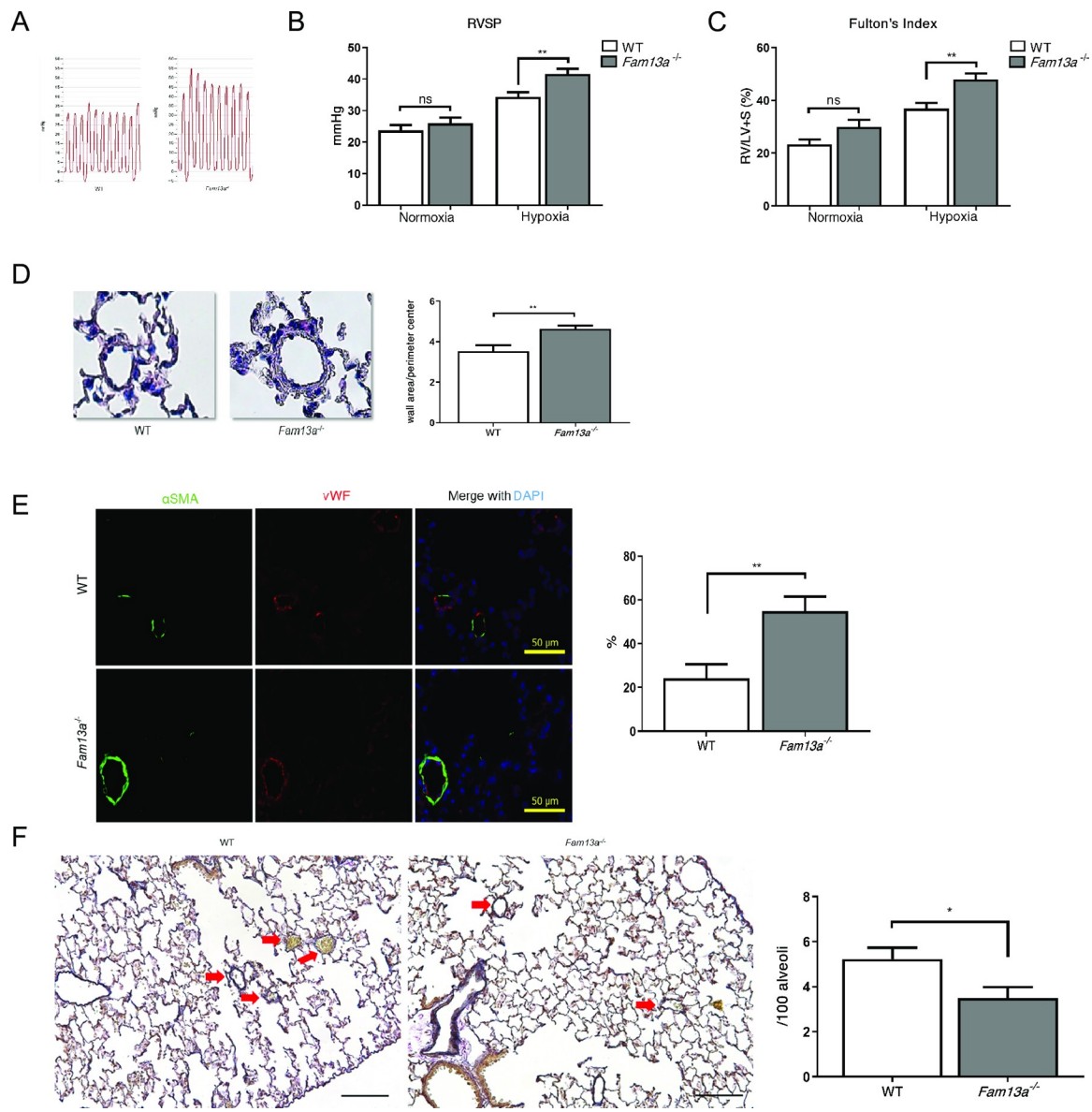

**Fig 3. Genetic loss of *FAM13A* exacerbates pulmonary hypertension.** (A) Pulse pressure diagram for right ventricles in WT and *Fam13a*[-/-] mice exposed to chronic hypoxia (10% $O_2$ for 3 weeks). (B) Right ventricular systolic pressure (RVSP) was measured in WT and *Fam13a*[-] mice exposed to either normoxic or hypoxic conditions (n = 5 each for normoxia group, n = 10 each for hypoxia group). (C) Ratio of right ventricle compared to left ventricle + septum (Fulton's Index) was calculated (n = 5 each for normoxia group, n = 10 each for hypoxia group). (D) HE staining of the lung sections in WT and *Fam13a*[-/-] mice exposed to chronic hypoxia. Prominent muscularization in small pulmonary artery was detected in the lungs of *Fam13a*[-/-] mice. Bars: 50 μm. Wall thickness was quantitatively analyzed (n = 8–9 each). (E) Immunohistochemistry for α-smooth muscle actin (α-SMA) and vWF in the lung sections of WT and *Fam13a*[-/-] mice exposed to chronic hypoxia. Muscularization of small pulmonary artery was deteriorated in the lungs of *Fam13a*[-/-] mice. Fully muscularized small pulmonary artery was quantified (n = 8–9 each). (F) Elastica van Gieson staining of the lung sections in WT and *Fam13a*[-] mice exposed to chronic hypoxia. Significant reduction in the number of small pulmonary artery (<50μm diameter) was observed in the lungs of *Fam13a*[-] mice. Average ratio of small pulmonary artery per 100 alveoli was calculated (n = 8–9 each). Data are presented as the mean ± SEM. *P < 0.05 and **P < 0.01.

*FAM13A* compared to the control cells after EndMT induction (Fig 5J). Considering a crucial role of β-catenin in promoting EndMT, *FAM13A* decelerates the EndMT process at least partially through inhibiting the β-catenin signaling.

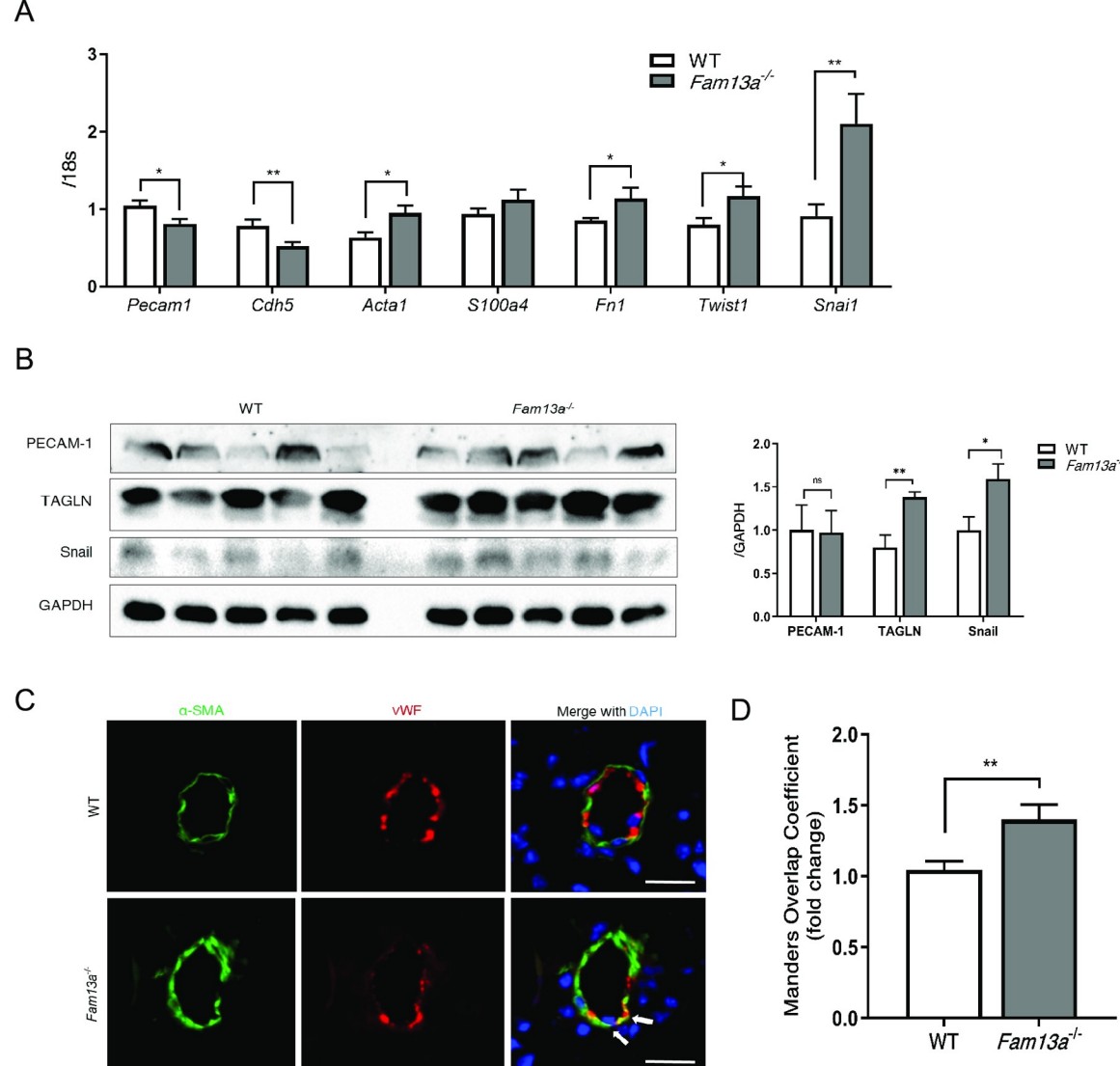

**Fig 4. Loss of *FAM13A* promotes EndMT.** (A) Quantitative real-time PCR of genes involved in the EndMT in the lungs isolated from WT and *Fam13a⁻/⁻* mice exposed to chronic hypoxia. Endothelial markers (*Pecam1* and *Cdh5*) expression was reduced, while mesenchymal markers (*Acta1* and *Fn1*) and their transcription factors (*Twist1* and *Snai1*) expression was enhanced in the lungs of *Fam13a⁻/⁻* mice (n = 9–11 each). (B) Immunoblotting for endothelial marker (PECAM-1) and mesenchymal markers (TAGLN and Snail) in the lungs isolated from WT and *Fam13a⁻/⁻* mice exposed to chronic hypoxia. TAGLN and Snail expression levels were significantly increased in the lung of *Fam13a⁻/⁻* mice, while PECAM-1 expression levels were similar between the groups (n = 5 each). (C) Immunohistochemistry for αSMA (mesenchymal marker) and vWF (endothelial marker) in the lungs. vWF-positive endothelial cells that are also positive for αSMA was more frequently detected in the lungs of *Fam13a⁻/⁻* mice. Arrows indicate the double-positive cells undergoing EndMT. Bars: 20μm. (D) Quantitative analysis for colocalization of mesenchymal and endothelial markers in pulmonary arteries assessed by the Manders overlap coefficient (n = 20 each). Data are presented as the mean ± SEM. *P < 0.05 and **P < 0.01.

## Discussion

Pulmonary arterial hypertension is a chronic and progressive disease that eventually leads to the right ventricular heart failure and premature death. Despite the recent progress in clinical treatment including the endothelin receptor antagonism and new class of vasodilator, there are still significant unmet clinical needs in the treatment of pulmonary hypertension. Identification of new and/or unknown pathways in the pathogenesis of pulmonary hypertension is

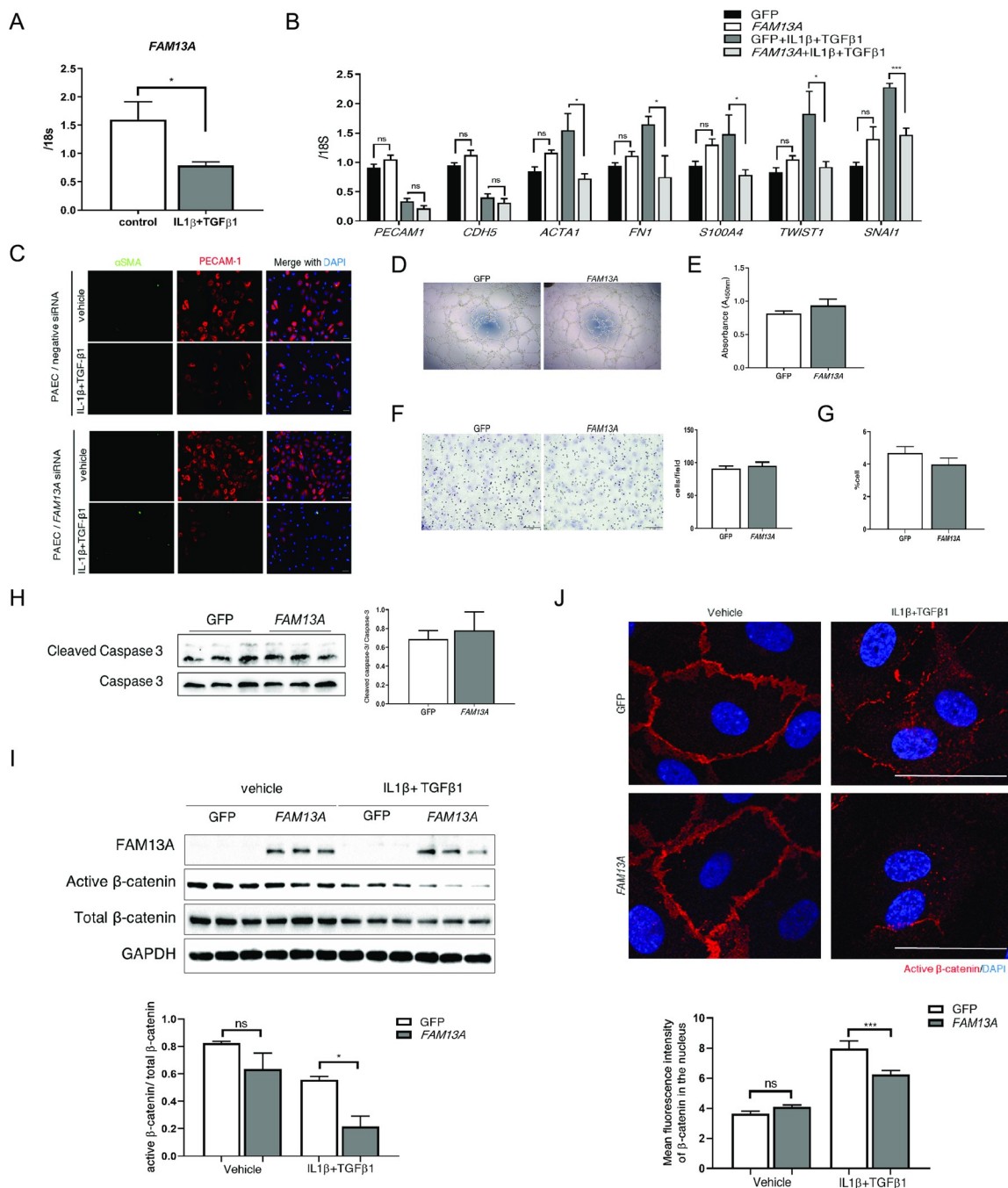

**Fig 5. _FAM13A_ decelerates EndMT through inhibiting β-catenin signaling.** (A) Quantitative real-time PCR of _FAM13A_ in PAECs. EndMT was induced by the treatment with TGF-β1 (10 ng/mL) and IL-1β (10 ng/mL) for 6 days. _FAM13A_ expression was reduced after EndMT induction in PAECs (n = 4 each). (B) Quantitative real-time PCR of genes involved in the EndMT in PAEC transfected with either GFP or _FAM13A_ in the presence or absence of EndMT induction. EndMT was induced by the treatment with TGF-β1 (10 ng/mL) and IL-1β (10 ng/mL) for 6 days. Overexpression of _FAM13A_ reduced mesenchymal markers and their transcription factors expression in PAECs after EndMT induction (n = 4–6 each). (C) Immunocytochemistry for endothelial (PECAM-1) and mesenchymal (αSMA) markers in PAECs transfected with either negative or Fam13a siRNA in the presence or absence of EndMT induction. (D) Tube-formation analysis in PAECs transfected with either GFP or _FAM13A_. (E) Proliferation assessed by WST-1 assay in PAECs transfected with either GFP or _FAM13A_ (n = 7 each). (F) Modified Boyden chamber assay in PAECs transfected with either GFP or _FAM13A_ (n = 3 each). (G, H) Apoptosis was induced by serum and growth factor depletion for 24 h in PAECs transfected with either GFP or _FAM13A_ (n = 3 each). Apoptosis was assessed by TUNEL-staining (G) and immunoblotting for cleaved caspase-3 (H). (I) Immunoblotting for _FAM13A_, active β-catenin, β-catenin, and GAPDH in PAECs transfected with either GFP or _FAM13A_ in the presence or absence of EndMT induction. Overexpression of _FAM13A_ reduced active β-catenin in PAECs after EndMT induction

(n = 3 each). (J) Immunohistochemistry for active β-catenin in PAECs transfected with either GFP or *FAM13A* in the presence or absence of EndMT induction. Bars: 50 μm. Nuclear accumulation of active β-catenin was quantified (n ≥ 50 nucleus/group). Data are presented as the mean ± SEM. *P < 0.05, **P < 0.01, and ***P < 0.001.

therefore important to improve the therapeutic strategies. In this manuscript, we revealed a previously undescribed role of *FAM13A* in the development of pulmonary hypertension. Given that *Fam13a* was reduced in the lungs of mice with pulmonary hypertension, and genetic loss of *FAM13A* exacerbated pulmonary hypertension, enhancing and/or preserving *FAM13A* in the lungs might have a therapeutic potential.

All forms of pulmonary arterial hypertension are characterized by vascular remodeling and dysfunction, of which mechanism includes multiple factors [2,3]. EndMT is one of the factors that cause abnormal vascular remodeling. EndMT is a cell transdifferentiation process in which endothelial cells lose endothelial specific markers and acquire mesenchymal properties. EndMT causes the loss of cell-cell adhesion, highly migratory and proliferative capacities in endothelial cells, leading to physiological and pathological cellular processes during embryonic development and disease onset. It has been reported that EndMT is involved in a variety of cardio-pulmonary diseases such as atherosclerosis [19], cardiac fibrosis [20], pulmonary fibrosis [21], and pulmonary hypertension [15,22,23]. In the remodeled vasculatures in pulmonary arterial hypertension, α-smooth muscle actin-expressing mesenchymal-like cells accumulate, especially in obstructive pulmonary vascular lesions. A fraction of these mesenchymal-like cells are derived from endothelial cells through the EndMT. Furthermore, alteration in BMPR-II signaling, which is critically involved in the pathogenesis of pulmonary hypertension, is linked to EndMT [22]. These findings indicate a crucial role of EndMT in pulmonary hypertension, and strongly suggest that EndMT is a promising therapeutic target.

Previous genome-wide association studies suggest a strong association between *FAM13A* and chronic lung diseases. *FAM13A* is expressed in various types of tissues and cells, including airway and alveolar epithelial cells in the lung, pulmonary vascular cells, and mature adipocytes in adipose tissue [10,11,24]. Accordingly, *FAM13A* has been involved in multiple biological processes such as epithelial cell regeneration, tumor cell proliferation and survival, and insulin signaling. *FAM13A* is known to harbor a Ras-homologous GTPase-activating protein (RhoGAP) domain that is important for proliferation and survival in lung adenocarcinoma cell A549 [24], and this domain could activates RhoA which can affect actin cytoskeleton and promotes epithelial-to-mesenchymal transition in cystic fibrosis lung [8]. *FAM13A* also has two coiled-coil domains that often play a role in protein-protein interactions. Indeed, *FAM13A* binds to insulin receptor substrate-1 in a coiled-coil domain-dependent manner, while it binds to protein phosphatase 2A (PP2A) independently of their coiled-coil domain [11]. Other study in COPD has revealed the importance of interaction between *FAM13A* and PP2A in bronchial epithelial cells that leads to β-catenin degradation through GSK3β-mediated phosphorylation, although the coiled-coil domain dependency is not clear [10].

In the current study, we have identified a protective role of *FAM13A* in the progression of pulmonary hypertension by utilizing mice in which *Fam13a* was genetically deleted. To our knowledge, this is the first report that identifies *Fam13a* expression in the lung vasculature. *FAM13A* has been reported to regulate the β-catenin signaling in airway epithelial cells [10,25], and we found that *FAM13A* negatively regulates β-catenin activity in endothelial cells as well. β-catenin signaling has been involved in epithelial-to-mesenchymal transition in pulmonary disease and cancer [8,12,26]. Also, β-catenin has been reported to promote EndMT through nuclear accumulation and subsequent activation of TCF/Lef transcription factors [13,14]. In the current study, we revealed that overexpression of *FAM13A* decelerates the

EndMT process in association with reduced active β-catenin levels and its nuclear accumulation in endothelial cells. These data strongly suggest that *FAM13A* decelerates EndMT process at least partially through inhibiting β-catenin signaling. It has been reported that β-catenin accumulation promotes survival and proliferation in PAECs through enhancing RhoA-Rac1 signaling [27]; however, we did not detect significant difference of angiogenic capacity in PAECs overexpressing *FAM13A*, despite the reduction in active β-catenin.

Because *FAM13A* is expressed in variety types of cells in the lungs, other *FAM13A*-mediated cellular processes might be involved in the pathogenesis of pulmonary hypertension. Nonetheless, our *in vivo* data using *Fam13a*$^{-/-}$ mice clearly showed that loss of *FAM13A* exacerbated pulmonary hypertension, and thus *FAM13A* is an attractive pharmacotherapeutic target for the treatment of pulmonary hypertension. However, these results have been shown in a hypoxia-induced mice model, where only medial lesions can develop, but not intimal lesions. Because pathological intimal lesion such as plexiform lesion is a hallmark of pulmonary arterial hypertension in human, further analyses to explore a role of endothelial *FAM13A* in the formation of intimal lesions are required to validate *FAM13A* as a feasible pharmacotherapeutic target for the treatment of pulmonary arterial hypertension.

## Supporting information

**S1 Table. Nucleotide sequence of primers.**
(PDF)

**S1 Fig. Original uncropped images for immunoblotting.**
(PDF)

## Author Contributions

**Conceptualization:** Koji Ikeda.

**Formal analysis:** Pranindya Rinastiti.

**Funding acquisition:** Noriaki Emoto.

**Investigation:** Pranindya Rinastiti, Elda Putri Rahardini, Kazuya Miyagawa, Naoki Tamada, Yuko Kuribayashi.

**Supervision:** Koji Ikeda, Ken-ichi Hirata, Noriaki Emoto.

**Writing – original draft:** Koji Ikeda.

**Writing – review & editing:** Pranindya Rinastiti, Noriaki Emoto.

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
