## [Decision Letter · Decision Letter 0]

27 Dec 2019

PONE-D-19-31771

Loss of family with sequence similarity 13, member A exacerbates pulmonary hypertension through accelerating endothelial-to-mesenchymal transition

PLOS ONE

Dear Dr. Ikeda,

Thank you for submitting your manuscript to PLOS ONE. After careful consideration, we feel that it has merit but does not fully meet PLOS ONE’s publication criteria as it currently stands. Therefore, we invite you to submit a revised version of the manuscript that addresses the points raised during the review process.

Although both reviewers were fairly enthusiastic about the manuscript, additional experiments will likely be needed to address the reviewers' comments, particularly the comments on apoptosis.

We would appreciate receiving your revised manuscript by Feb 10 2020 11:59PM. To enhance the reproducibility of your results, we recommend that if applicable you deposit your laboratory protocols in protocols.io, where a protocol can be assigned its own identifier (DOI) such that it can be cited independently in the future. For instructions see: http://journals.plos.org/plosone/s/submission-guidelines#loc-laboratory-protocols

We look forward to receiving your revised manuscript.

Kind regards,

James West, PhD

Academic Editor

PLOS ONE

Journal Requirements:

4. To comply with PLOS ONE submissions requirements, please provide methods of sacrifice in the Methods section of your manuscript.

Reviewers' comments:

Reviewer's Responses to Questions

**Comments to the Author**

1. Is the manuscript technically sound, and do the data support the conclusions?

Reviewer #1: Yes

Reviewer #2: Yes

2. Has the statistical analysis been performed appropriately and rigorously? 

Reviewer #1: Yes

Reviewer #2: Yes

3. Have the authors made all data underlying the findings in their manuscript fully available?

Reviewer #1: Yes

Reviewer #2: Yes

4. Is the manuscript presented in an intelligible fashion and written in standard English?

Reviewer #1: Yes

Reviewer #2: Yes

5. Review Comments to the Author

Reviewer #1: In their submitted paper for PLOS ONE “Loss of family with sequence similarity 13, member A exacerbates pulmonary hypertension through accelerating endothelial-to-mesenchymal transition” Rinastiti P and co-worker described a protective role of FAM13A in the development of pulmonary hypertension through decelerating EndMT in a hypoxia-induced mouse model. The authors also suggested that controlling FAM13A expression might be a new therapeutic target in PH. This study is supposed to be interesting and the development of a new therapeutic strategy seems to be essential for the future treatment of pulmonary hypertensive diseases.

However, major questions arise and there are some concerns that you have to handle.

Major comments:

1) Why did the authors choose hypoxia-induced pulmonary hypertension model, instead monocrotaline-induced or SU5416/Hypoxia-induced model?

2) In this study, the authors demonstrated that FAM13A have a protective role in the development of pulmonary arterial remodeling through inhibiting beta-catein signaling. However, these results have been shown only in a hypoxia-induced mice model, where only medial lesions can develop, not intimal lesions. When FAM13A is considered as an attractive pharmacotherapeutic target for the treatment of pulmonary ARTERIAL hypertension, the authors need to think about strategies for intimal lesions including complex vascular lesions. I think that the authors should discuss on it in discussion session.

Minor comments:

1) In Materials and Methods, Line 83, Right ventricular � right ventricular

2) In Figure 1, did 1C and 1D indicate findings of the lung from hypoxia-induced mice?

3) In Figure 4, the authors mentioned that arrows indicate the double-positive cells undergoing EndMT. However, there should be no arrow!

Reviewer #2: In this manuscript, the authors demonstrate that FAM13A is reduced in the lungs from mice with pulmonary hypertension, and the loss of FAM13A exacerbates the development and progression of pulmonary hypertension. FAM13A decelerates EndMT process at least partially through inhibiting beta- catenin signaling. Overall the experiments are well designed and the results support the conclusions draw by the authors. However, this reviewer has some concerns listed below,

1. In figure 4, WB should be done and quantified to show the loss of FAM13A affect the EndMT.

2. It would be better to do immunostaining to show overexpression of FAM13A induce EndMT through IL-1beta and TGFbeta.

3. In Figure 5F, more specific experiments are required to confirm the apoptosis assay.

6. PLOS authors have the option to publish the peer review history of their article (what does this mean?). If published, this will include your full peer review and any attached files.

Reviewer #1: No

Reviewer #2: No

---

## [Author Response · Author response to Decision Letter 0]

6 Jan 2020

Responses to the Reviewers’ comments

Reviewer #1: 

In their submitted paper for PLOS ONE “Loss of family with sequence similarity 13, member A exacerbates pulmonary hypertension through accelerating endothelial-to-mesenchymal transition” Rinastiti P and co-worker described a protective role of FAM13A in the development of pulmonary hypertension through decelerating EndMT in a hypoxia-induced mouse model. The authors also suggested that controlling FAM13A expression might be a new therapeutic target in PH. This study is supposed to be interesting and the development of a new therapeutic strategy seems to be essential for the future treatment of pulmonary hypertensive diseases.

However, major questions arise and there are some concerns that you have to handle.

Major comments:

Comment-1

Why did the authors choose hypoxia-induced pulmonary hypertension model, instead monocrotaline-induced or SU5416/Hypoxia-induced model?

Response-1

Thank you for the comment. As mentioned by the Reviewer, monocrotaline- or SU5416/Hypoxia-induced pulmonary hypertension model shows more severe phenotypes with higher pulmonary arterial pressure and worsened pulmonary artery remodeling than hypoxia-induced pulmonary hypertension model in rats. Especially, SU5416/Hypoxia-induced pulmonary hypertension model develops occlusive neointimal arteriopathy that is similar to plexiform lesion, a histological hallmark of human pulmonary arterial hypertension. However, both monocrotaline- and SU5416/Hypoxia-induced pulmonary hypertension models have been established in rats, but not in mice.

In mice, the classic model of pulmonary hypertension is the hypoxia-induced pulmonary hypertension model in which mice develop mild pulmonary hypertension in association with mild to moderate muscularization of small pulmonary arteries. In order to use Fam13a-deficient mice to explore a role of Fam13a in pulmonary hypertension in vivo, we chose the well-established hypoxia-induced pulmonary hypertension model in this study.

Comment-2

In this study, the authors demonstrated that FAM13A have a protective role in the development of pulmonary arterial remodeling through inhibiting beta-catein signaling. However, these results have been shown only in a hypoxia-induced mice model, where only medial lesions can develop, not intimal lesions. When FAM13A is considered as an attractive pharmacotherapeutic target for the treatment of pulmonary ARTERIAL hypertension, the authors need to think about strategies for intimal lesions including complex vascular lesions. I think that the authors should discuss on it in discussion session.

Response-2

Thank you for the comments. We sincerely agree with the Reviewer’s comments, and described the limitation of our study as follows.

“However, these results have been shown in a hypoxia-induced mice model, where only medial lesions can develop, but not intimal lesions. Because pathological intimal lesion such as plexiform lesion is a hallmark of pulmonary arterial hypertension in human, further analyses to explore a role of endothelial FAM13A in the formation of intimal lesions are required to validate FAM13A as a feasible pharmacotherapeutic target for the treatment of pulmonary arterial hypertension” (page 28, Line 435-440).

Minor comments:

Comment-3

In Materials and Methods, Line 83, Right ventricular is right ventricular.

Response-3

Thank you for the comment. We have corrected the uppercase “R” into the lowercase “r”.

Comment-4

In Figure 1, did 1C and 1D indicate findings of the lung from hypoxia-induced mice?

Response-4

Thank you for the comment. These data were obtained using the lungs isolated from WT and Fam13a-deficient mice under normoxia condition. We have described this important experimental condition in the revised manuscript.

Comment-5

In Figure 4, the authors mentioned that arrows indicate the double-positive cells undergoing EndMT. However, there should be no arrow!

Response-5

Thank you for the comment. We have added the arrows indicating the double-positive cells in the new Figure 4C.

 

Reviewer #2: 

In this manuscript, the authors demonstrate that FAM13A is reduced in the lungs from mice with pulmonary hypertension, and the loss of FAM13A exacerbates the development and progression of pulmonary hypertension. FAM13A decelerates EndMT process at least partially through inhibiting beta- catenin signaling. Overall the experiments are well designed and the results support the conclusions draw by the authors. However, this reviewer has some concerns listed below,

Comment-1

In figure 4, WB should be done and quantified to show the loss of FAM13A affect the EndMT.

Response-1

Thank you for the comment. We have newly analyzed the EndMT through WB for PECAM-1(endothelial marker), and transgelin (TAGLN) and Snail (mesenchymal marker) in the lungs of WT and Fam13a-deficient mice exposed to chronic hypoxia. As shown in the new Figure 4B, TAGLN and Snail protein levels were enhanced in the lung of Fam13a-deficient mice comparing to those in WT mice, while PECAM-1 expression levels were not different between the groups. No difference in PECAM-1 expression levels might be due to its expression in hematopoietic cells in addition to its expression in endothelial cells. Nevertheless, enhanced TAGLN and Snail protein expression levels in the lung of Fam13a-deficient mice further support a role of Fam13a in the EndMT process.

Comment-2

It would be better to do immunostaining to show overexpression of FAM13A induce EndMT through IL-1beta and TGFbeta.

Response-2

Thank you for the comment. According to the comment, we have performed immnostaining for aSMA (mesenchymal marker) and PECAM-1 (endothelial marker) using PAECs transfected with either negative or Fam13a siRNA in the presence or absence of IL-1b+TGF-b treatment. As shown in the new Figure 5C, PECAM-1-positive cells decreased after the EndMT-induction, and the loss of PECAM-1 expression appeared to be enhanced in PAEC transfected with Fam13a siRNA as compared to that in PAEC transfected with negative siRNA. On the other hand, we could not detect the aSMA expression by immunostaining in both groups even after the treatment with IL-1b+TGF-b. This is probably because of the insufficient sensitivity of the antibody to detect aSMA expression in PAEC undergoing EndMT. These new data further support an inhibitory role of Fam13a in the EndMT process.

Comment-3

In Figure 5F, more specific experiments are required to confirm the apoptosis assay.

Response-3

Thank you for the comment. We newly analyzed the apoptosis by using TUNLE-staining and immunoblotting for cleaved caspase-3, instead of the Hoechst nuclear staining. As shown in the new Figure 5G, apoptosis was not affected by Fam13a-overexpression in PAEC, as was suggested by the previous Hoechst nuclear staining.

---

## [Decision Letter · Decision Letter 1]

27 Jan 2020

Loss of family with sequence similarity 13, member A exacerbates pulmonary hypertension through accelerating endothelial-to-mesenchymal transition

PONE-D-19-31771R1

Dear Dr. Ikeda,

We are pleased to inform you that your manuscript has been judged scientifically suitable for publication and will be formally accepted for publication once it complies with all outstanding technical requirements.

With kind regards,

James West, PhD

Academic Editor

PLOS ONE

Additional Editor Comments (optional):

Reviewers' comments:

Reviewer's Responses to Questions

**Comments to the Author**

1. If the authors have adequately addressed your comments raised in a previous round of review and you feel that this manuscript is now acceptable for publication, you may indicate that here to bypass the “Comments to the Author” section, enter your conflict of interest statement in the “Confidential to Editor” section, and submit your "Accept" recommendation.

Reviewer #1: All comments have been addressed

Reviewer #2: All comments have been addressed

2. Is the manuscript technically sound, and do the data support the conclusions?

Reviewer #1: Yes

Reviewer #2: Yes

3. Has the statistical analysis been performed appropriately and rigorously? 

Reviewer #1: Yes

Reviewer #2: Yes

4. Have the authors made all data underlying the findings in their manuscript fully available?

Reviewer #1: Yes

Reviewer #2: Yes

5. Is the manuscript presented in an intelligible fashion and written in standard English?

Reviewer #1: Yes

Reviewer #2: Yes

6. Review Comments to the Author

Reviewer #1: All comments that I raised have been properly addressed. This manuscript is now acceptable for publication.

Reviewer #2: Thank the authors for their continued work. To this point, I am very satisfied with this revision which has improved the quality of the paper. Congrats to them for their efforts.

7. PLOS authors have the option to publish the peer review history of their article (what does this mean?). If published, this will include your full peer review and any attached files.

Reviewer #1: Yes: Seiichiro Sakao

Reviewer #2: No

---

## [Editor Report · Acceptance letter]

30 Jan 2020

PONE-D-19-31771R1 

Loss of family with sequence similarity 13, member A exacerbates pulmonary hypertension through accelerating endothelial-to-mesenchymal transition 

Dear Dr. Ikeda:

I am pleased to inform you that your manuscript has been deemed suitable for publication in PLOS ONE. Congratulations! Your manuscript is now with our production department. 

With kind regards,

on behalf of

Dr. James West 

Academic Editor

PLOS ONE